# Photoluminescence Modulation of Ruddlesden-Popper Perovskite via Phase Distribution Regulation

**DOI:** 10.3390/nano13030571

**Published:** 2023-01-31

**Authors:** Xinwei Zhao, Ting Zheng, Weiwei Zhao, Yuanfang Yu, Wenhui Wang, Zhenhua Ni

**Affiliations:** School of Physics, Southeast University, Nanjing 211189, China

**Keywords:** Ruddlesden-Popper Perovskite, photoluminescence, phase regulation, immediate state

## Abstract

The intrinsic chaotic phase distribution in Ruddlesden-Popper Perovskite (RPP) hinders its further improvement of photoluminescence (PL) emission and limits its application in optical devices. In this work, we achieve the phase distribution regulation of RPP by varying the composition ratio of organic bulky spacer cations 1-naphthylmethylamine (NMA) and phenylethyl-ammonium (PEA), which is controllable and nondestructive for structures of RPP. By suppressing the small n-phase, the PL intensity emission of RPP is further improved. Through the time-resolved PL (TRPL) measurements, we find the PL lifetime of the sample with 66% PEA concentration increases with the temperature initially and possesses the highest values of $\tau_{1}$ and $\tau_{2}$ at ~255 K, indicating the immediate state assisting exciton radiative recombination, and it can be modulated by phase manipulation in RPP. The immediate state may outcompete other non-radiative decay channels for excited carriers, leading to the PL enhancement in RPP, and broadening its further application.

## 1. Introduction

Ruddlesden-Popper Perovskite (RPP), with chemical formula ${A^{\prime }}_{2}A_{n-1}B_{n}X_{3n+1}$ (where A’ is bulky organic spacer cations, A is an organic or inorganic cation, B is ${\mathrm{Pb}}^{2+}$, X I s halide anion) [1,2], has received a lot of attention in recent years. In RPP, A’ and ABX_3_ with different dielectric constants are stacked alternately to form a quantum well-like structure [3]. In theory, this structure greatly reduces the non-radiative recombination of excitons and produces high photoluminescence quantum yields (PLQY) [4,5,6]. However, the chaotic phase in the thin film caused by varying values of n in ${A^{\prime }}_{2}A_{n-1}B_{n}X_{3n+1}$ samples leads to irregular quantum well widths, limiting further enhancement of its PLQY [7,8,9]. Therefore, modulating the values of n in RPP to obtain unity phase components is an effective method for enhancing the PLQY and achieving further photonic applications of RPP [10]. Efforts have been made to control the phase distribution during the spin-coating process and alter the values of n in RPP films, such as changing the components of A and A’ cations [11], using additional additives [12] and adding anti-solvent [13]. However, these methods can result in weak stability, additional defects or control issues [13,14,15] and restrict the implementation of PL enhancement in practical applications.

In this work, we were able to gradually modify the phase components of RPP by altering the proportion of phenylethylammonium (PEA) and 1-naphthylmethylamine (NMA). By effectively suppressing the n phase, we were able to further improve the PLQY of RPP and address the challenges faced by the traditional methods. The results of TRPL showed that PL lifetime significantly increased at low temperatures, indicating the presence of an immediate state. The immediate state-assisted radiative recombination outcompetes other non-radiative recombination, such as the defect-related recombination. Moreover, the efficiency of the immediate state is improved by regulating the n phase. This work presents an effective method for enhancing the PL intensity in RPP through phase regulation and provides a foundation for the advanced practical application of RPP, such as a light emitter.

## 2. Materials and Methods

### 2.1. Preparation of the ${\mathrm{PEA}}_{2}{\mathrm{FA}}_{n-1}{\mathrm{Pb}}_{n}{\mathrm{Br}}_{3n+1}$ and ${\mathrm{NMA}}_{2}{\mathrm{FA}}_{n-1}{\mathrm{Pb}}_{n}{\mathrm{Br}}_{3n+1}$

The solution containing 0.4 mol/L ${\mathrm{PEA}}_{2}{\mathrm{FA}}_{n-1}{\mathrm{Pb}}_{n}{\mathrm{Br}}_{3n+1}$ and ${\mathrm{NMA}}_{2}{\mathrm{FA}}_{n-1}{\mathrm{Pb}}_{n}{\mathrm{Br}}_{3n+1}$ was obtained by mixing FABr, NMABr or PEABr and PbBr (Xi’an Polymer Light Technology Corp.) in N, N-dimethylformamide (Shanghai Aladdin Biochemical Technology Corp.) at a ratio of 1:4:4 and stirring it at 60 °C for 12 h. The two well-stirred solutions were then combined to prepare the solutions with a PEA content of 20%, 33%, 50%, 66% and 80%. The original two solutions were used as solutions with a PEA content of 0% and 100%, resulting in a total of seven solutions with different PEA concentrations. Subsequently, these solutions were spin coated onto SiO_2_ substrates (3000 rpm, 30 s) that had been cleaned with acetone, methanol, deionized water, O_2_ plasma treated and baked for 20 min at 70 °C on a hot plate. All preparations were carried out in a N_2_ glove box. (The process diagram of the sample preparation is illustrated in Appendix A.)

### 2.2. Characterizations

All samples were evaluated at the RT. XRD patterns and the Absorption spectrum were obtained using a Smart X-ray diffractometer (Smartlab, Rikagu) and HITACHI U-3900 UV−vis spectrophotometer, respectively. Steady-state PL spectra were collected using the Confocal Raman Spectrometer (LabRAM HR UV-Visible, Horiba Jobin Yvon), with the excitation from a 325 nm He-Cd laser (KIMMON). TRPL measurements were performed using a home-made setup, where a 405 nm pulsed laser doubled by an fs-laser source (Chameleon Compact OPO, center wavelength: 810 nm, repetition rate: 80 MHz), a BBO crystal, was used for the excitation. The laser beam (P = 31 μW) was focused by a microscopy objective (50×, NA: 0.5,) to collect the signal. The AFM mappings were imaged using Dimension ICON (Bruker).

## 3. Results and Discussion

Seven samples of PEA/NMA mixed RPP film samples were prepared in the same condition, with PEA concentrations of 0%, 20%, 33%, 50%, 66%, 80% and 100%. The structure of PEA/NMA mixed RPP is shown in Figure 1a, along with the corresponding optical and SEM images in Figure 1b. Surface morphology of the mixed bulky spacer cation RPP film, as seen in the Figure 1c, suggests that the prepared RPP is a well-crystallized polycrystalline film with average grain size of 84 nm and high surface roughness (RMS ≈ 7.36 nm) [16]. As shown in Figure 1d, the samples with different PEA exhibit evident absorption at ~ 414 nm, ~445 nm and ~525 nm. The former two peaks are from the interband absorption in the small n-phases (n = 2 and 3) [17], while the peak at ~525 nm is from the interband absorption in large n-phases that are close to the bulk phase [13]. The changes in the absorbance peak intensity suggest that the effective phase composition adjustment of perovskite film is achieved through the change in the organic bulky spacer cation mixing ratio [17].

The phase composition change was further investigated using X-ray diffraction (XRD) patterns (Figure 1e). The strong diffraction peaks at 14.8° and 29.9° correspond to the (100) and (200) diffraction patterns of the 3D perovskite FAPbBr_3_ and are attributed to the large n-phase component in RPP [18]. The diffraction peaks at 12.4° (002) and 37.4° (211) are from the small n-phase component in RPP [19]. The normalized intensity of the peak shows that the phase distribution orientation in the RPP tends to be parallel to the substrate [20]. As shown in Figure 1f, the (100) and (200) diffraction peak intensity of the large n phase increases with the increase in the PEA concentration, while the diffraction peak of the large n phase remains constant. This trend suggests a gradual change in the phase composition change towards the unity large n-phase.

The PL spectra of samples with different PEA concentration are shown in Figure 2a. All the PL data are collected under same condition (Excitation wavelength: 325 nm, power: 6 μW). It is evident that the PL intensity of the mixed A’ cation RPP film is efficiently enhanced. According to the fitting results of peak intensity shown in Figure 2b, the PL intensity initially shows an increasing trend with increasing PEA concentration, reaching the maximum intensity at ~66% concentration. This enhancement indicates that the PL control of the RPP film by mixing organic bulky spacer cations is realized. As the photoluminescence yield of perovskite is relatively high, the additional enhancement is worthy of special research. Furthermore, the PL peak position of the mixed bulky organic cations RPP red-shifts from 537 nm to 541 nm with increasing PEA concentrations (Figure 2c), which can be attributed to the change in the phase composition of RPP [21]. Due to the energy funnel effect of RP perovskite with fewer organic components, the final PL peaks are all corresponding to the large n phase, and the full width at half maximum (FWHM) of the PL spectrum remain constant at different PEA concentrations (Appendix A).

To gain a deeper understanding of the mechanism behind the PL enhancement by adjusting the PEA concentration, we performed TRPL measurement on samples with different PEA concentrations, as shown in Figure 3a. Bi-exponential functions (I=A1e−ττ1+A2e−ττ2) were used to fit all the TRPL data. Here, I and τ are the PL intensity and PL decay time, respectively. The terms $\tau_{1}$ and $\tau_{2}$ describe the fast and slow components of the decay, with the relative weights of A_1_ and A_2_. The changes of these lifetimes with different PEA concentrations are shown in the Figure 3b. As the ratio of mixed PEA increases,${\;\tau }_{1}$ initially increases from 0.51 ns to 1.22 ns first, before decreasing to 0.55 ns. Similarly, $\tau_{2}$ increases from 1.6 ns to 3.4 ns first, before decreasing to 2.0 ns. According to the time scales of the two processes, we attribute the $\tau_{1}$ and $\tau_{2}$ to the carrier lifetime during the PL process at the surface and bulk of the RPP thin film, respectively [22]. The highest values of $\tau_{1}$ and $\tau_{2}$ are both located at 66% PEA concentration sample.

The increasing PL intensity in Figure 3a,b in conjunction with the lifetime suggests the suppression of non-radiative recombination. As defects play a significant role in non-radiative recombination in perovskite, we conducted temperature-dependence TRPL of samples with 66% and 0% PEA concentration to comprehend the underlying mechanism of PL enhancement (Figure 3 and Appendix A). As shown in 3d, the $\tau_{2}$ shows an initial increase, up to an order of magnitude, and decreases subsequently while $\tau_{1}$ remains almost constant. Moreover, the weight of $\tau_{2}$ tends to be 100% at peak (Appendix A), indicating the significance of this slow decay channel for excited carries recombination. Similar phenomena emerge at the 0 % PEA RPP sample with different peak temperature and weight (Figure 3e and Appendix A). The variation in $\tau_{2}$ with increasing temperature excludes the trap-mediated non-radiative recombination [23] (Appendix A) and suggests that the existence of immediate states assisted non-radiative to radiative recombination [24], which can be further confirmed by other techniques. It should be noted that the $\tau_{2}$ of 66% PEA concentration sample has a larger timescale and presents higher weight.

A possible PL enhancement mechanism is illustrated in Figure 3f, in which these immediate states store several carriers (process ①), then activated by energy injection to complete the carrier transition (process ②) and photon emission (process ③). Evidently, this process prolongs the PL lifetime and increases overall PL intensity. In our case, the activation energy comes from thermal disturbance and ~ 255 K is the optimal working temperature in 66% PEA concentration RPP, while ~ 250 K is for 0% (Figure 3e). However, defect-related nonradiative recombination will mainly lead to $\tau_{2}$ decreasing above 255 K, rather than other bimolecular recombination of free electrons and holes or thermal effect [23,25], as confirmed in Appendix A. Also, due to the high thermal stability of RP perovskite, the PL is less affected by thermal effect [26,27]. This can be summarized as follows: 1. The immediate states-assisted radiative recombination and defect-related nonradiative recombination are the two main competitive channels that synergistically determine the carrier decay in this work; 2. The phase regulation effectively affects the varied optimal temperature for immediate states-assisted radiative recombination, thus affecting the PLQY.

## 4. Conclusions

In conclusion, we have presented a new method for enhancing the PLQY by altering the ratio of organic bulky spacer cations in RPP films. Comparing the conventional methods, the phase modulation by changing composition in this work is more controllable and nondestructive for the structures of RPP. Using TRPL, we have found the lifetime of the sample increases with the temperature, suggesting the presence of intermediate states. Moreover, the different phase composition resulting from various PEA concentrations can regulate the immediate states of RPP, which assists radiative recombination. Additionally, the working status for immediate states-assisted PL can be adjusted through phase distribution, providing a new avenue for further PLQY enhancement.

## Figures and Tables

**Figure 1 nanomaterials-13-00571-f001:**
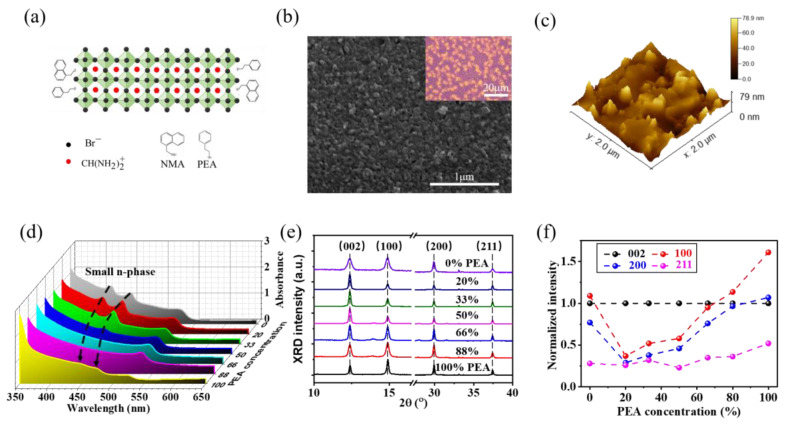
Phase distribution regulation of RPP (at the room temperature). (**a**) Schematic diagram of mixed A’ cation RPP structure. (**b**) Optical and SEM images of RPP samples (66% PEA). (**c**) AFM image of RPP film (66% PEA), surface roughness ≈ 7.36 nm. (**d**) Absorbance spectra of mixed organic bulky spacer cations RPP films. (**e**) XRD patterns of mixed organic bulky spacer cations RPP films. The diffraction peaks at 14.8° (100) and 29.9° (200) are from large n-phase, and the diffraction peak at 12.4° (002) and 37.4° (211) is from small n-phase. (**f**) Normalized intensity of diffraction peaks as a function of PEA ratio.

**Figure 2 nanomaterials-13-00571-f002:**
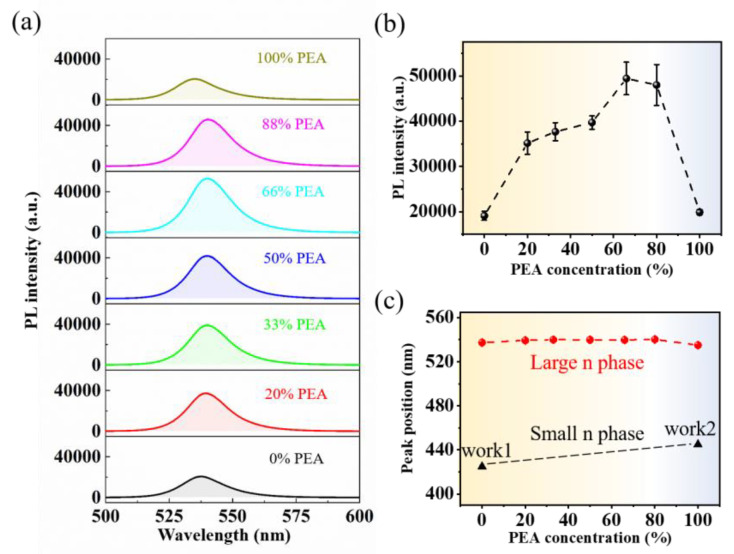
(**a**) PL spectra of mixed organic bulky spacer cations RPP films (at room temperature). (**b**) The correlation between the PL intensity and the PEA concentration in the mixed organic bulky spacer cations. (**c**) The correlation between the PL peak position and the PEA concentration [13,18].

**Figure 3 nanomaterials-13-00571-f003:**
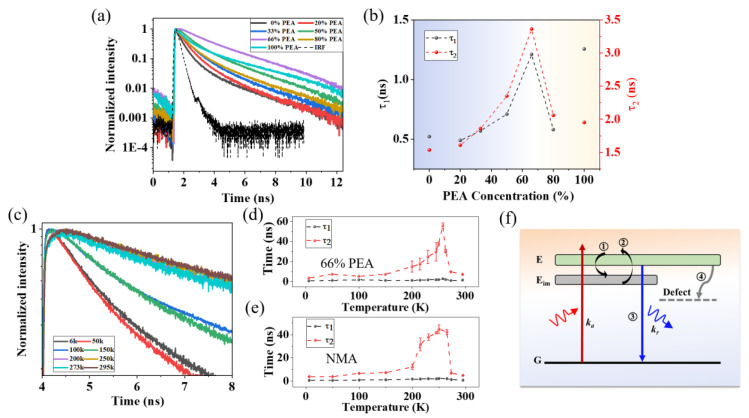
(**a**) TRPL spectra of mixed organic bulky spacer cations RPP films under the same 33uw light excitation. (**b**) $\tau_{1}$ and $\tau_{2}$ processes as a function of PEA components. (**c**) TRPL spectra of 66% PEA sample with different temperature. (**d**) $\tau_{1}$ and $\tau_{2}$ processes in 66% PEA samples as a function of temperature. (**e**) $\tau_{1}$ and $\tau_{2}$ processes in single NMA samples as a function of temperature. (**f**) Schematic illustration of the PL enhancement mechanism of RPP Perovskite by mixed bulky spacer cations.

## Data Availability

The data that support the findings of this study are available from the first author or corresponding authors upon reasonable request.

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
