# Peer review of "Photoluminescence Modulation of Ruddlesden-Popper Perovskite via Phase Distribution Regulation"

_nanomaterials, 2023, doi:10.3390/nano13030571_

Round 1

Reviewer 1 Report

The authors propose a new method for enhancing the PL quantum yield. Thie approach is based on changing the 166 mixed ration of organic bulky spacer cations in RPP films. I found the study very interesting and timely. Most importantly, the concept is fully demonstrated by performing experiment and providing the measurment results. The manuscript is nicely written and well organized. In addition, the obtained results seem scientifically sound. Given these points, I recommend the publication of the work. 

Author Response

Dear reviewer,

Thank you very much for taking the time to review this manuscript. I truly appreciate your thoughtful consideration and positive feedback on this manuscript.

Best Regards,

Zhenhua Ni,

School of Physics

Southeast University, Nanjing, China, 211189

Email address: [email protected]

Reviewer 2 Report

The authors in the manuscript entitled "Photoluminescence Modulation of Ruddlesden -Popper Perovskite (RPP) via Phase Distribution Regulation" have tried to show the increase in photoluminescence of RPP by the small n-phase suppression. The manuscript can be of interest to the readership.  There are several flaws in the manuscript as described below:

1. The introduction section needs to be modified completely. It does not relate to the advancement of works in this field and the significance of their work. 

2. In figure 1, the absorption spectra are hard to understand as absorbance data is not clear.   Could you present it in another way such that absorbance can be distinguished? 

3. The authors do not talk about the basis of the phase distribution (lines 45-55). Were the proportions taken randomly? 

4. The preparation method of PEA2FAn-1PbnBr3n+1 and NMA2FAn-1PbnBr3n+1 seems hard to follow by a third researcher. Could you present it more legibly? Maybe in a set of cartoon images. 

5. There are so many grammatical errors in the manuscript. For instance, in the conclusion section: a new mean (line 166) should be replaced with a new means/method,  ration (line 167) should be replaced with ratio, by TRPL (line 169) should be replaced with from TRPL, we acquire (line 169) we find, etc. Extensive English language style modification is needed.  

Author Response

Please see the attachemnt.

Reviewer 3 Report

Your paper entitled, “Photoluminescence Modulation of Ruddlesden -Popper Perovskite via Phase Distribution Regulation,” is interesting.  Developers and users of optoelectronic devices (LEDs, photo and laser diodes, solar cells, etc.) can benefit from Ruddlesden -Popper Perovskite (RPP) thin films with enhanced photoluminescence quantum yield (PLQY). Your advanced approach to enhancing the PLQY by changing the mixed ratio of organic bulky spacer cations in RPP thin films and suppressing the n phase diminishes the effect of intrinsic chaotic phase distribution that suppresses improvement of the photoluminescence (PL) emission. A gradual modulation of phase components of RPP was achieved by varying the mixing ratio of phenylethylammonium (PEA) and 1-naphthylmethylamine (NMA).

 My comments/recommendations are provided below.

  • Reference Line 13 – Recommend that TRPL (time-resolved photoluminescence) be defined in the Abstract.
  • Reference Lines 34 and 35 – The authors state that the drawbacks in conventional methods include weak stability, additional defects, or control issues. Recommend that a statement be provided in the Abstract and Conclusion on how your approach solved these issues.
  • Recommend that the authors identify the best practical application for the advanced technology disclosed in this paper.

I made a few suggestions to improve sentence structure (See the attached PDF document – Comments / Recommendations are in Sticky Notes).

Round 2

Reviewer 2 Report

Thank you. Looks good.